# Ethical considerations related to drone use for environment and health research: A scoping review protocol

Remy Hoek Spaans[1,2]☯ *, Bruna Drumond[3,4]☯, Kim Robin van Daalen[1], Ana Claudia Rorato Vitor[5], Alison Derbyshire[2], Adriano Da Silva[6], Raquel Martins Lana[1], Mauricio Santos Vega[7], Gabriel Carrasco-Escobar[8], Maria Isabel Sobral Escada[5], Claudia Codeço[9], Rachel Lowe[1,10,11] *

1 Barcelona Supercomputing Center (BSC), Barcelona, Spain, 2 Liverpool School of Tropical Medicine (LSTM), Liverpool, United Kingdom, 3 Programa de Pós-Graduação em Saúde Pública, Escola Nacional de Saúde Pública Sergio Arouca (ENSP), Fundação Oswaldo Cruz (Fiocruz), Rio de Janeiro, Brazil, 4 Programa Institucional Territórios Sustentáveis e Saudáveis (PITSS), Vice-Presidência de Ambiente, Atenção e Promoção da Saúde (VPAAPS), Fundação Oswaldo Cruz (Fiocruz), Rio de Janeiro, Brazil, 5 National Institute for Space Research (INPE), Laboratory for Investigation in Socio-Environmental Systems (LiSS), São José dos Campos, Brazil, 6 Instituto de Comunicação e Informação Científica e Tecnológica em Saúde/Fiocruz (Icict/Fiocuz), Rio de Janeiro, Brazil, 7 Grupo en Biologia Matematica y Computacional, Departamento de Ciencias Biologicas, Universidad de los Andes, Bogota, Colombia, 8 Health Innovation Laboratory, Institute of Tropical Medicine "Alexander von Humboldt", Universidad Peruana Cayetano Heredia, Lima, Peru, 9 Programa de Computação Científica, Fundação Oswaldo Cruz (Fiocruz), Rio de Janeiro, Brazil, 10 Catalan Institution for Research and Advanced Studies (ICREA), Barcelona, Spain, 11 Centre on Climate Change & Planetary Health and Centre for Mathematical Modelling of Infectious Diseases, London School of Hygiene & Tropical Medicine, London, United Kingdom

☯ These authors contributed equally to this work.
* remy.hoekspaans@bsc.es (RHS); rachel.lowe@bsc.es (RL)

**Data Availability Statement:** No datasets were generated or analysed during the current study. All relevant data from this study will be made available upon study completion.

## Abstract

### Introduction

The use of drones in environment and health research is a relatively new phenomenon. A principal research activity drones are used for is environmental monitoring, which can raise concerns in local communities. Existing ethical guidance for researchers is often not specific to drone technology and practices vary between research settings. Therefore, this scoping review aims to gather the evidence available on ethical considerations surrounding drone use as perceived by local communities, ethical considerations reported on by researchers implementing drone research, and published ethical guidance related to drone deployment.

### Methods and analysis

This scoping review will follow the Preferred Reporting Items for Systematic Reviews and Meta-Analyses extension for scoping reviews (PRISMA-ScR) and the Joanna Briggs Institute (JBI) guidelines. The literature search will be conducted using academic databases and grey literature sources. After pilot testing the inclusion criteria and data extraction tool, two researchers will double-screen and then chart available evidence independently. A content analysis will be carried out to identify patterns of categories or terms used to describe ethical considerations related to drone usage for environmental monitoring in the literature using

**Funding:** RL is the principal investigator of the Wellcome Trust (https://wellcome.org/) funded HARMONIZE project. The project award reference is 224694/Z/21/Z. The funders had no role in the study design.

**Competing interests:** The authors have declared that no competing interests exist.

**Abbreviations:** AACODS, Authority, Accuracy, Coverage, Objectivity, Date, Significance; BASE, Bielefeld Academic Search Engine; BSC, Barcelona Supercomputing Center; CASP, Critical Appraisal Skills Programme; ELSA, ethical, legal and social aspects; ELSI, ethical, legal and social issues; FIOCRUZ, Fundação Oswaldo Cruz; ICREA, Catalan Institution for Research and Advanced Studies; JBI, Joanna Briggs Institute; LSTM, Liverpool School of Tropical Medicine; MEDLINE, Medical Literature Analysis and Retrieval System Online; MeSH, Medical Subject Headings; NGO, non-governmental organisations; OSF, Open Science Framework; PCC, Population, Concept, Context; PRESS, Peer Review of Electronic Search Strategies; PRISMA, Preferred Reporting Items for Systematic Reviews and Meta-Analyses; PRISMA-ScR, Preferred Reporting Items for Systematic Reviews and Meta-Analyses extension for scoping reviews; RPAS, remotely piloted aerial systems; RRI, Responsible Research and Innovation; UAS, unmanned aerial system; UAV, unmanned aerial vehicle; UNICEF, United Nations Children's Fund; VHL, Virtual Health Library; WoS, Web of Science.

the R Package RQDA. Discrepancies in any phase of the project will be solved through consensus between the two reviewers. If consensus cannot be reached, a third arbitrator will be consulted.

## Ethics and dissemination

Ethical approval is not required; only secondary data will be used. This protocol is registered on the Open Science Framework (osf.io/a78et). The results will be disseminated through publication in a scientific journal and will be used to inform drone field campaigns in the Wellcome Trust funded HARMONIZE project. HARMONIZE aims to develop cost-effective and reproducible digital infrastructure for stakeholders in climate change hotspots in Latin America & the Caribbean and will use drone technology to collect data on fine scale landscape changes.

## Introduction

Drones (definition in S1 File), also referred to as unmanned aerial vehicles (UAVs) or remotely piloted aerial systems (RPAS), have been taken up by civil society at an exponential rate over the past decade due to their high mobility, low costs, and high endurance for multiple tasks. Whilst initially developed for military use, drones have since been adopted for agriculture, commerce, humanitarian aid and disaster response, and more recently public health research. Within research applications, drone use can be roughly divided into two main categories: transportation/delivery and environmental monitoring (definition in S1 File). An example of transport/delivery is a study examining the feasibility of transporting medicine or blood products to remote health facilities [1]. Another example is using drones to deliver interventions for vector control (e.g., the release of *Wolbachia*-carrying mosquitoes or sterile insects, adulticide spraying, and larvicide delivery) [2, 3]. These activities are characterised by linear flights between points and moving objects into or out of a location. Contrastingly, environmental monitoring does not involve any physical interaction with the environment but remotely records information about a phenomenon, area, or research subject. This includes activities such as recording multispectral imagery, taking air samples, and recording meteorological variables [4, 5]. Flights are recorded over a study area by manually controlling the drone or programming pre-planned flights. Depending on the research aim, the same area may be covered repeatedly. This can raise concerns among local communities residing in the area.

Ethical considerations can differ according to the perspectives of the stakeholders involved in the research process. We identify three main types of stakeholders: local communities, researchers, and institutions responsible for upholding ethical standards (universities, funders, non-governmental organisations [NGOs]). In this scoping review, we consider these perspectives to synthesise evidence on the ethical considerations in drone use for environment and health research. A definition of ethical considerations for the purpose of this scoping review can be found in the S1 File.

What the research community defines as ethical considerations within environmental and health research are rooted in the four bioethical principles of 1) respect for autonomy, 2) non-maleficence, 3) beneficence, and 4) justice proposed and continuously developed by Beauchamp and Childress [6]. Since then, other frameworks to analyse aspects related to ethics have been produced to accommodate new scientific discoveries and ways of thinking. The development of the ethical, legal and social aspects (ELSA) or issues (ELSI) framework as part

of the Human Genome Project, was adapted to include considerations specific to emerging technologies and identifies eleven domains (S1 File) [7]. The Responsible Research and Innovation (RRI) is broader than the more evaluative approach of ELSI and has a greater focus on the interaction between science and society. RRI can be described as "a transparent, interactive process by which societal actors and innovators become mutually responsive to each other with a view to the (ethical) acceptability, sustainability and societal desirability of the innovation process and its marketable products (to allow a proper embedding of scientific and technological advances in our society)" [7, 8]. The RRI is implemented considering 5 key areas: 1) gender equality, 2) open access, 3) citizen engagement, 4) science education, and 5) ethics. Citizen engagement and ethics are of particular interest in the context of drone use.

Although ethical frameworks have been used to analyse evidence gathered on drone use within healthcare and humanitarian settings, the same has not been done for drone use within environmental and health research [9, 10]. Cawthorne *et al.*, in their review on the use of drones in a healthcare setting, used the four bioethical principles and added a fifth principle from the field of artificial intelligence; explicability [11]. Wang *et al.* based the analysis in their scoping review for drone use in humanitarian settings on the ELSI principles (S1 File) which they incorporated in a value-sensitivity framework for the University of Zurich´s ethical guidelines for humanitarian drone use in collaboration with the Red Cross [12]. To our best knowledge, there is no unified set of publicly available guidelines for the use of drones for environment and health research and narrative, scoping or systematic reviews have been published on this subject to date.

## Rationale

There is a need to describe evidence available on current research ethical practices regarding interactions between researchers and local communities, on how drone use is perceived by these communities, as well as existing ethical guidelines to identify best practices and ethical concerns that have remained unaddressed in research using drones for environmental monitoring. Only the application of drones for environmental monitoring will be considered as this evokes different interactions with local communities and consequently different ethical considerations than drone use for transport/delivery. The results of this work can support researchers planning or currently conducting studies using drones for environmental monitoring, ethics committee members, and research institutions or NGOs aiming to provide ethical guidance.

## Research aim

**Research aim.** Summarise all evidence available on ethical considerations surrounding drone use within environmental and health research.

This will be addressed by summarising three types of evidence, that correspond to the stakeholders' perspectives, through the following sub-questions:

1. What are the perceptions, experiences and views of local communities related to ethical drone use within environmental monitoring? (qualitative studies and questionnaires)

2. What are the ethical practices currently described by researchers using drones for environmental monitoring, especially relating to their interactions with local communities? (case or implementation studies)

3. What ethical guidelines exist to inform the design and implementation of studies using drones for environmental monitoring? (ethical guidelines)

## Materials and methods

### PCC

**Population.**   1) Local communities inhabiting research areas where drones are used for environmental monitoring. 2) Researchers conducting drones for environmental monitoring. 3) Institutions issuing ethical guidance within environmental and health research.

**Concept.**   Ethical practice relating to the interaction between drone research activities and local communities. Specifically, local communities' perceptions, experiences, and opinions concerning the use of drones in research in their territories.

**Context.**   Using drones for environmental monitoring within environment and health research in populated areas globally. Any stage of the research process reported on or investigated by researchers or institutions issuing guidance that pertains to ethical conduct, with community engagement activities of particular interest.

### Search strategy

The academic literature search will be conducted using the following databases: EBSCO Medical Literature Analysis and Retrieval System Online (MEDLINE) Complete, Scopus, Web of Science (WoS), Global Health Database, and the Virtual Health Library (VHL) Regional Portal. A primary search is developed for EBSCO MEDLINE Complete in collaboration with an information specialist (AD), based on three core concepts—"drone(s)", "environmental monitoring" and "ethical considerations" or "community perceptions"—described in Table 1 and S1 File and informed by Wang *et al.*´s search strategy [9]. A combination of free-text terms and Medical Subject Headings (MeSH) will be used. Following the Peer Review of Electronic Search Strategies (PRESS) 2015 guidelines, the search will be peer-reviewed by an information specialist [13] and subsequently adapted for the other databases. Reference lists of relevant articles will be checked to identify further literature meeting the inclusion criteria and Google Scholar will be used to carry out forward and backward citation searching. No time restrictions will be applied to the search to avoid missing any relevant literature: all articles from the inception of the searched databases to the date of the literature search will be included. If full-text

**Table 1. Core concepts and search terms for the primary search strategy developed for EBSCO MEDLINE complete.**

| Core concept | Search terms |
|---|---|
| Drone | (drone* OR UAV* OR UAS* OR Unmanned Aerial Devices (MH)) OR ((unmanned OR uncrewed) AND ("aerial vehicle*" OR "air vehicle*" OR "aerial system*" OR "aerial device*" OR aircraft)) |
| Environmental monitoring | "remote* sens*" OR data OR image* OR photograph* OR map* OR surveillance OR survey* OR monitor* OR Remote Sensing Technology (MH) OR Environmental Monitoring (MH) OR Environmental Indicators (MH) OR Data Collection (MH) OR Population Surveillance (MH) |
| Community perceptions or ethical considerations | (community OR public) AND (perception* OR aware* OR engag* OR participat* OR accept* OR involve* OR concern OR cooperat* OR support* OR response* OR view* OR consent OR compliance)) OR ethic* OR "social impact*" OR "societal impact" OR privacy OR confidential* OR Ethics (MH) OR Community Participation (MH) OR Cooperative Behaviour (MH) OR Informed Consent (MH) OR Privacy (MH) OR Confidentiality (MH) OR Research Subjects (MH) OR "code of conduct" OR "code of ethics" OR "ethics code" |

(MH) = Medical Subject Heading (MeSH), a controlled and hierarchically organized vocabulary produced by the National Library of Medicine.

**Table 2. Template for data table to report retrieved search results.**

| Database | Search terms | Number of papers | Date |
|---|---|---|---|
| | | | |

articles are unavailable, the corresponding authors will be contacted to request access via e-mail or ResearchGate within the next two months. All non-English records will be reviewed by research team members with reading literacy in several languages; Dutch, English, French, German, Spanish, Portuguese or translated (using Google translate). In cases where this is not sufficient, the authors will reach out to their extended research networks.

For each database, the search terms, number of papers retrieved, and date of collection will be reported, as depicted in Table 2.

The authors will undertake searches of the grey literature using Google Advanced Search and the grey literature databases such as the Bielefeld Academic Search Engine (BASE). Search terms will be iteratively adapted from Table 1, PDF and Word file types will be targeted and only the first 5 pages of results are included. After the general screening, using a method similar to forward citation searching, a list of websites and organisations that are deemed to do relevant work involving drones in a health or environment research setting will be compiled. This will NGOs, international organisations, universities, aviation authorities, research funders, and research institutions. Examples include United Nations Children's Fund (UNICEF), International Committee of the Red Cross, University of Zurich, Civil Aviation Authorities, and World Health Organisation. Using Google Advanced Search, the web domains of these organisations will be searched. Theses repositories will not be searched, but non-published theses will be included if they show up through the grey literature search and fit the inclusion criteria.

## Inclusion and exclusion criteria

Table 3 describes the inclusion and exclusion criteria that will be used for abstract and title and full-text screening.

Pilot testing of the inclusion and exclusion criteria will be conducted by: 1) double-screening a random sample of 50 records (25 academic and 25 grey literature), and 2) assessing resulting conflicts and clarifying inclusion and exclusion criteria if deemed necessary. This process will be repeated until an inter-rater reliability of 90% is reached.

## Evidence selection

After removing duplicates within Zotero, records are imported into Rayyan (https://rayyan.ai/) and double-screened by two independent researchers [14]. One researcher is Brazilian and based at the Oswaldo Cruz Foundation (Fiocruz) with experience in Geography, Environment and Health research, whilst the other is Dutch and based at the Barcelona Supercomputing Center (BSC) in Spain with field experience using drones in Malawi.

The first screening round will be based on titles and abstracts following the inclusion and exclusion criteria. The second screening round will focus on the full-texts. For grey literature, the first round is based on the title and summary (where available) or the first page of the document. After completion of the round, results are compared, and conflicts are discussed until a consensus between the two researchers is reached. If a consensus cannot be reached a third arbitrator will make the decision.

**Table 3. Inclusion and exclusion criteria.**

| | Inclusion criteria | Exclusion criteria |
|---|---|---|
| 1 | Drones used for research purposes | Recreational, commercial, military, or humanitarian use |
| 2 | Focus of the study is on drones to collect environmental monitoring data | Drones were used for other purposes such as transport, tracking, etc. |
| 3 | Location of drones deployed involves human populations | Studies implemented in uninhabited areas |
| 4 | Document mentions interactions between researchers and communities and/or ethical considerations, concerns, or views related to drone use | No interactions between researchers and the community are described or ethical considerations are not discussed |
| 5 | Academic literature including: quantitative studies (e.g., cross-sectional, cohort, case-control), qualitative studies (e.g., action research, ethnographic, phenomenological, case), mixed-method studies, perspective papers | Conference abstracts or studies with no full-text available (after reaching out to the corresponding author by email or Researchgate) |
| 6 | Grey literature issued by research institutions, research funding agencies, regional, local and national governmental bodies, multilateral organisations, and non-governmental organisations, theses | Forms of grey literature such as blogs, podcasts, news stories, social media posts |
| 7 | Most current version of the document | Document was a draft, has not been officially released, or has been replaced with another document published later |
| 8 | Primary research articles and originally written grey literature. | Studies with a secondary study design that reported on data already included in another study. For example, systematic reviews and other summaries or synthesis of primary articles and originally written pieces. |
| 9 | Ethical guidelines collated in such a way that they comprise of a unique perspective by the primary author with original details (even if based on previously constructed principles) | Ethical guidelines that are duplicated from another original source (indicated by reference, or time of publication). |
| 10 | Records in any language. If the original language is not English, the record will be assigned to a native speaker within the team. If no native speaker is available, online translation services will be used. | If translations are ambiguous or assessed to be of poor quality, the record will be excluded. |

The number of full-text sources excluded and reasons for exclusion are tracked and reported through a flowchart of the review process, in line with the Preferred Reporting Items for Systematic Reviews and Meta-Analyses (PRISMA) guidelines [15].

## Data charting

We expect to retrieve studies and reports that include both qualitative and quantitative data, as well as ethical guidelines. Whereas some records will likely have ethical practices and/or considerations as the direct research topic, we also expect to retrieve studies that use drones and only partially describe their ethical considerations and practices. It is also expected that the degree to which drones are the focus will differ between ethical guidelines. Quality assessment will be conducted for studies with a qualitative study design using the Critical Appraisal Skills Programme (CASP) Qualitative Checklist [16]. No formal quality assessment will be done for record types for which the study was not designed to investigate ethical drone use (e.g., case studies designed to measure image classification accuracy). Ethical guidance identified through the grey literature search will be assessed through the Authority, Accuracy, Coverage, Objectivity, Date, Significance (AACOD) checklist [17]. Quality assessment will be done by

**Table 4. Data extraction form.**

| Variable | Format |
|---|---|
| Bibliographic information | Strings describing: author names, year of publication, title, country of origin of the authors, journal etc. |
| Type of literature | Categorical: journal article, pre-prints, thesis, dissertation, report, guideline, etc. |
| Study design | Categorical: qualitative studies (e.g., case study, action research, ethnographic, grounded theory), quantitative studies (e.g., cross-sectional, prospective or retrospective cohort, case control), other |
| Dominant perspective of the source | Categorical: researcher, local community, institution issuing guidance |
| Study aims and objectives | String |
| Characteristics of the study population (e.g., age, sex) in which the drone is deployed | String |
| Geographical location in which the research was conducted | String recoded to categorical (region, country) |
| Land use and cover | Categorical: agricultural / built up / grassland / savanna / rainforest, etc. |
| Type of data collected by drone | String recoded to categorical based on results |
| Drone model | String recoded to categorical based on results |
| Flight plan | String recoded to categorical based on results (altitude, coverage, pattern, planned/manual etc.) |
| Dates the study was conducted, and follow-up period if applicable | Dates |
| Main outcome of interest | String |
| Main finding(s) within the context of this scoping review´s objectives | String of text describing main findings regarding ethical research practices and community engagement strategies; community perceptions |
| If ethical approval was sought | Binary: yes/no |
| Type of interaction with the community | String recoded to categorical: interviews, focus groups, other types of participatory methodologies and community engagement strategies |
| Ethics-related theories or frameworks referenced in the text | String |
| Ethical considerations mentioned | String recoded to categorical based on results |

one of the reviewers and checked by the second reviewer. Any discrepancies in judgement will be discussed until consensus is reached.

Data will be independently extracted by two researchers from all eligible evidence. An extraction sheet will be jointly developed, and pilot tested by two researchers on 5 records. Iterative improvements are made until both reviewers agree the tool captures the information well.

The information to be extracted from each included record (where relevant and available) is described in Table 4.

Disagreements will be resolved through consensus between two researchers. If a consensus is not reached, a third arbiter will be involved.

## Analysis of the evidence and presentation of the results

Data summaries, maps and graphs will be produced for quantitative results for each main type of evidence i) community perceptions, ii) current ethical research practices with a focus on

community engagement, and iii) ethical guidelines, based on the results from the data extraction tool.

A content analysis will be conducted to identify patterns of categories or terms used to describe ethical considerations related to drone use for environmental monitoring in the literature. The content analysis will synthesise themes around i) community perceptions, ii) current ethical research and community engagement practices, and iii) ethical guidelines concerning the use of drones for environmental monitoring in environment and health research. We will also identify the theoretical approaches from the field of ethics used in these studies. Two researchers will independently use inductive analysis to generate and agree on a codebook. The R Package RQDA version 0.2–8 will be used to code the qualitative evidence into categories and format it for further analysis [18]. The results will be presented in a narrative synthesis, supplemented with thematic maps, tables, and graphs with descriptive statistics on included studies and their outcome. All sections of the published results will follow the Preferred Reporting Items for Systematic Reviews and Meta-Analyses extension for scoping reviews (PRISMA-ScR), which can be found in the S2 File [15].

## Ethics and dissemination

Ethical approval is not required for this study, no original patient or participant data will be collected. The authors plan on publishing the Zotero repository of the search and screening results via the Open Science Framework (OSF). The results of the study will be published in a peer-reviewed scientific journal.

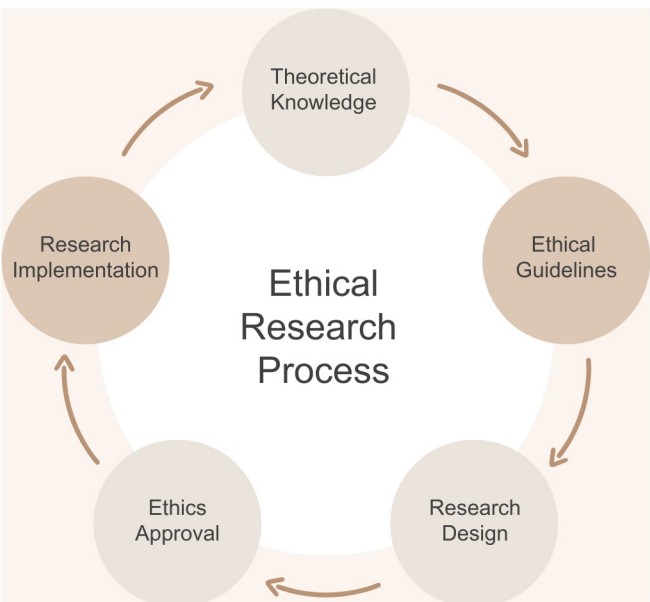

**Fig 1. Five stages of the iterative research process during which ethical considerations are relevant.** The process through which ethical guidance within the research community is created and updated. In the context of drone use for remote sampling, this entails 1) creating a theoretical understanding of drone technology within research and corresponding ethical considerations, 2) the creation of ethical guidelines by institutions, which are then 3) applied by researchers when designing a new study, followed by 4) an assessment of the study by an ethical committee for approval, after which 5) the study is implemented in line with appropriate ethical practice which could involve mention of specific community engagement activities. When new ethical questions, information, or best practices are identified during the study, they will then contribute to further understanding of 1) theoretical ethical considerations.

### Changes to the protocol

The heterogeneous nature of the evidence base may cause a need to adapt the methodology as new themes or sources of evidence emerge. All deviations and refinements made to the protocol will be reported in the published scoping review.

## Discussion

While ethical dilemmas may arise during each of the research stages (Fig 1), not all information arising from these processes is documented or made publicly available. It is important to reflect on this bias when interpreting and discussing the results from the scoping review. In this scoping review, we expect to retrieve most of our results from on the Research Implementation stage described in Fig 1, as well as mapping which Ethical Guidelines incorporate drone use and are publicly available. Information emerging from the Research Implementation stage will likely include data from qualitative studies directly giving insight into community perceptions or that have ethical considerations as the research topic. Data from (likely quantitative) studies that do not have research ethics as the direct topic, but that make use of drones for environmental monitoring and describe interactions with the community or ethical considerations are also of interest. This approach will enable us to compare and reflect on publicly available guidelines in the context of findings from original research reporting on the experiences of communities and researchers involved in drone research.

## Supporting information

**S1 File. Definitions of the key concepts and ELSI principles used by Wang *et al*..**
(DOCX)

**S2 File. PRISMA-ScR checklist.**
(DOCX)

## Author Contributions

**Conceptualization:** Remy Hoek Spaans, Bruna Drumond, Claudia Codeço, Rachel Lowe.

**Funding acquisition:** Mauricio Santos Vega, Gabriel Carrasco-Escobar, Claudia Codeço, Rachel Lowe.

**Methodology:** Remy Hoek Spaans, Bruna Drumond, Kim Robin van Daalen, Alison Derbyshire, Adriano Da Silva, Claudia Codeço, Rachel Lowe.

**Supervision:** Claudia Codeço, Rachel Lowe.

**Visualization:** Remy Hoek Spaans.

**Writing – original draft:** Remy Hoek Spaans, Bruna Drumond.

**Writing – review & editing:** Kim Robin van Daalen, Ana Claudia Rorato Vitor, Raquel Martins Lana, Mauricio Santos Vega, Gabriel Carrasco-Escobar, Maria Isabel Sobral Escada, Claudia Codeço, Rachel Lowe.

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
