## [Decision Letter · Decision Letter 0]

3 Nov 2023

PONE-D-23-15407Ethical considerations related to drone use for environment and health research: a scoping review protocol

PLOS ONE

Dear Dr. Hoek Spaans,

Thank you for submitting your manuscript to PLOS ONE. After careful consideration, we feel that it has merit but does not fully meet PLOS ONE’s publication criteria as it currently stands. Therefore, we invite you to submit a revised version of the manuscript that addresses the points raised during the review process.

We look forward to receiving your revised manuscript.

Kind regards,

Sathishkumar Veerappampalayam Easwaramoorthy

Academic Editor

PLOS ONE

Journal Requirements:

Reviewers' comments:

Reviewer's Responses to Questions

**Comments to the Author**

1. Does the manuscript provide a valid rationale for the proposed study, with clearly identified and justified research questions?

Reviewer #1: Yes

Reviewer #2: Yes

2. Is the protocol technically sound and planned in a manner that will lead to a meaningful outcome and allow testing the stated hypotheses?

Reviewer #1: Partly

Reviewer #2: Yes

3. Is the methodology feasible and described in sufficient detail to allow the work to be replicable?

Reviewer #1: Yes

Reviewer #2: Yes

4. Have the authors described where all data underlying the findings will be made available when the study is complete?

Reviewer #1: Yes

Reviewer #2: Yes

5. Is the manuscript presented in an intelligible fashion and written in standard English?

Reviewer #1: Yes

Reviewer #2: Yes

6. Review Comments to the Author

You may also provide optional suggestions and comments to authors that they might find helpful in planning their study.

Reviewer #1: The subject is very interesting with many applications to the clinical practice.

I have some remarks.

I consider that the introduction is quite extensive and a significant part could be the discussion where could be the whole fact of the subject and perhaps any considerations about the protocol.

Reviewer #2: I congratulate the authors for a relevant topic selection and researching an unexplored area. I have minor feedback, please find it in the list below:

Abstract

1. Line 30-32: The title should correspond to the primary objective (currently it is in secondary objectives, as people's perceptions may not just be about ethics, please specify the primary objective specific to ethics). Please edit.

2. Line 34: Please edit the spelling of 'Joanna', can abbreviate as JBI

3. Line 39-40: Please use future tense

4. Line 39-40: A suggestion is to add 'consensus between the two reviewers' or 'disagreements will be resolved by a senior reviewer'

5. Please state full form of abbreviations the first time they appear in text

Introduction

1. Line 69-77: Please consider removing repeated information.

2. Line 125: Do the authors mean "publicly available guidelines" or scoping review or both? (the above few sentences are review examples)

3. Please try to shorten the introduction, it can be more focussed towards the questions.

4. Please try to strengthen the rationale, a suggestion is to insert some text from introduction to rationale or have no delineation between these sections

5. Line 158: It is a suggestion to shift PCC to the methods section. Please edit the context to geographic location (global or high income countries?) or setting or interests

6. Line 161: Please see the comment about objectives in the abstract section comments

Methods:

1. Line 175 & 177: EBSCO and MEDLINE should be in capital case because they are abbreviations.

2. Line 182-183: A suggestion is to read about PRISMA-S checklist to find its suitability for your review. Please compare PRESS and PRISMA-S and propose the more suitable one.

3. Line 185-187: Is there any reason why the authors are including articles from date of inception of database?

4. Line 188: Is there any time period that you will wait for the research authors to reply to your request?

5. Line 190-191: Google translate can be used during screening, but sometimes Google translate does not give accurate translation. Please pilot it before using.

6. Line 228: Please edit the spelling of Rayyan

7. Line 254-280: Request to shift the data to a table.

8. Please write about quality assessment (will be carried out or not?)

9. Please consider (not mandatory) stakeholder engagement in the review

General

1. Please define ethical decisionmaking and research areas in S1 file

2. Please include a list of abbreviations

7. PLOS authors have the option to publish the peer review history of their article (what does this mean?). If published, this will include your full peer review and any attached files.

Reviewer #1: **Yes: **Dr Ioannis Vogiatzis

Reviewer #2: **Yes: **Prachi Pundir

---

## [Author Response · Author response to Decision Letter 0]

29 Nov 2023

Dear reviewers,

I would like to thank you for taking the time to review our manuscript, your thoughtful remarks, and your expressed interest in the topic. 

Both reviewers commented on the length of the introduction and I will therefore respond to both reviewers on this topic first. 

Reviewer 1 offered part of the solution: to move part of the introduction to the discussion. I have moved the section and figure discussing which parts of the research process would be captured by the results of our scoping review to the discussion at the end of the protocol. On reflection, this section is about publication bias, which will help with the interpretation of results and is therefore better suited for the discussion section. 

Reviewer 2 pointed out that there was some repetition on the subject of previous reviews done with a focus on humanitarian drone use. The repeated information has been removed. Reviewer 2 also commented on the coherence between the study rationale, main aim, and introduction. Because it is recommended in the JBI guideline for the structuring of a scoping review protocol, I have kept the rationale section. It may also help readers who are scanning the protocol to skip to the rationale section without having to read the full introduction. However, some sections of the introduction have been moved to the rationale section, which hopefully improves clarity and flow from introduction to rationale. 

Overall, due to moving and deleting sections, the introduction is around a page shorter. Hopefully, this meets your expectations and has made the protocol easier to read. As a consequence of shortening the introduction I had to adjust the numbering of the references and reference list. Two co-authors have been added to the list after joining the HARMONIZE because of their valued expertise using drones in the field. 

Please find responses to each individual point raised on the next page. Responses are formatted in red, for clarity. 

Kind regards,

Remy Hoek Spaans

 

Response to the editor

Some small formatting issues have been corrected throughout the manuscript. File names of the uploaded files have been changed according to your request. 

Response to Reviewer 1:

The subject is very interesting with many applications to the clinical practice.

I have some remarks.

I consider that the introduction is quite extensive and a significant part could be the discussion where could be the whole fact of the subject and perhaps any considerations about the protocol.

The suggested edits have been made and were addressed in the general letter. 

Response to Reviewer 2:

I congratulate the authors for a relevant topic selection and researching an unexplored area. I have minor feedback, please find it in the list below:

Abstract

1. Line 30-32: The title should correspond to the primary objective (currently it is in secondary objectives, as people's perceptions may not just be about ethics, please specify the primary objective specific to ethics). Please edit.

Edits have been made in the text to emphasize the ethical aspect of the work. The amended sentence now reads:

“Therefore, this scoping review aims to gather the evidence available on ethical considerations surrounding drone use as perceived by local communities, ethical considerations reported on by researchers implementing drone research, and published ethical guidance related to drone deployment.”

2. Line 34: Please edit the spelling of 'Joanna', can abbreviate as JBI

Proposed edit made from “Joana” to “Joanna”. 

3. Line 39-40: Please use future tense

Proposed edit made, sentence now reads;

“Discrepancies in any phase of the project will be solved through consensus between the two reviewers. If consensus cannot be reached, a third arbitrator will be consulted.”

4. Line 39-40: A suggestion is to add 'consensus between the two reviewers' or 'disagreements will be resolved by a senior reviewer'

Proposed edit made.

“Discrepancies in any phase of the project will be solved through consensus between the two reviewers. If consensus cannot be reached, a third arbitrator will be consulted.”

5. Please state full form of abbreviations the first time they appear in text

The text has been read through again, and abbreviations have mentioned in full when first mentioned. All abbreviations have also been added to the abbreviation list in accordance with general comment 2. 

Introduction

1. Line 69-77: Please consider removing repeated information.

Proposed deletion made, considering that the scoping review by Wang et al. is already discussed in other paragraphs further on in the document.

2. Line 125: Do the authors mean "publicly available guidelines" or scoping review or both? (the above few sentences are review examples)

Both, edited in the text for clarity. The sentence now reads:

“To our best knowledge, there is no unified set of publicly available guidelines for the use of drones for environment and health research and narrative, scoping or systematic reviews have been published on this subject to date.”

3. Please try to shorten the introduction, it can be more focussed towards the questions.

The suggestion was taken on board and has been addressed in the general section at the start of this document. 

4. Please try to strengthen the rationale, a suggestion is to insert some text from introduction to rationale or have no delineation between these sections

The suggestion was taken on board and has been addressed in the general section at the start of this document. 

5. Line 158: It is a suggestion to shift PCC to the methods section. Please edit the context to geographic location (global or high income countries?) or setting or interests

Suggestion accepted. We are interested to find out how much research has been done on this globally. The only requirement is that the area is populated since we are interested in interactions between researchers and communities. 

6. Line 161: Please see the comment about objectives in the abstract section comments

The objectives have been edited to emphasize the ethical aspect of the work:

1. What are the perceptions, experiences and views of local communities related to ethical drone use within environmental monitoring? (qualitative studies and questionnaires)

2. What are the ethical practices currently described by researchers using drones for environmental monitoring, especially relating to their interactions with local communities? (case or implementation studies)

3. What ethical guidelines exist to inform the design and implementation of studies using drones for environmental monitoring? (ethical guidelines)

Methods:

1. Line 175 & 177: EBSCO and MEDLINE should be in capital case because they are abbreviations.

Correction made.

2. Line 182-183: A suggestion is to read about PRISMA-S checklist to find its suitability for your review. Please compare PRESS and PRISMA-S and propose the more suitable one.

PRESS focuses specifically on the search strategy, while PRISMA-ScR focuses on the overall review protocol). PRESS is a tool used by secondary librarians to peer-review the search strategy.

The adoption of the PRESS guidelines is recommended by the PRISMA-ScR guidelines: 

"Additional details to report include the person who did the literature search (for example, an experienced librarian or information specialist) and whether it was peer-reviewed by another librarian using the Peer Review of Electronic Search Strategies (PRESS) checklist, a set of recommendations for librarians and other information specialists to use when evaluating electronic search strategies" - Appendix of the PRISMA-ScR

Tricco, Andrea C., et al. "PRISMA extension for scoping reviews (PRISMA-ScR): checklist and explanation." Annals of internal medicine 169.7 (2018): 467-473.

A small change has been made in the text to make it more concise and avoid repetition. 

A combination of free-text terms and Medical Subject Headings (MeSH) will be used. Following the Peer Review of Electronic Search Strategies (PRESS) 2015 guidelines, the search will be peer-reviewed by an information specialist (14) and subsequently adapted for the other databases.

3. Line 185-187: Is there any reason why the authors are including articles from date of inception of database?

We wanted to avoid missing any relevant articles. After inspecting the bar chart showing the number of search results over time, it appeared that most publications are within the last ten years. Since no earlier scoping reviews have been conducted, to see the number of publications over time might be of interest in and of itself. 

An edit has been made in the text to explain:

“No time restrictions will be applied to the search to avoid missing any relevant literature: all articles from the inception of the searched databases to the date of the literature search will be included.”

4. Line 188: Is there any time period that you will wait for the research authors to reply to your request?

We will wait 2 months, edit made in text. 

“If full-text articles are unavailable, the corresponding authors will be contacted to request access via e-mail or ResearchGate within the next two months.”

5. Line 190-191: Google translate can be used during screening, but sometimes Google translate does not give accurate translation. Please pilot it before using.

Our expectation is that besides English the most common other languages will be Portuguese or Spanish, of which we have native speakers in the team. Based on preliminary results we do not anticipate language to be a major barrier. The authors combined have a large extended international network of researchers that speak that speak Mandarin, Arabic, Farsi, Russian for example. 

The text now reads:

“All non-English records will be reviewed by research team members with reading literacy in several languages; Dutch, English, French, German, Spanish, Portuguese or translated (using Google translate).In cases where this is not sufficient, the authors will reach out to their extended research networks.” 

6. Line 228: Please edit the spelling of Rayyan

Correction made.

7. Line 254-280: Request to shift the data to a table.

Requested edit made, the information is now available in table format.

8. Please write about quality assessment (will be carried out or not?)

We make a distinction between articles that do and do not have a study design to measure what we are interested in (ethical values and behaviour surrounding drones).

Addition made in text: 

"Quality assessment will be conducted for studies with a qualitative study design using the Critical Appraisal Skills Programme (CASP) Qualitative Checklist. No formal quality assessment will be done for record types for which the study was not designed to investigate ethical drone use (e.g., case studies designed to measure image classification accuracy). Ethical guidance identified through the grey literature search will be assessed through the Authority, Accuracy, Coverage, Objectivity, Date, Significance (AACODS) checklist. Quality assessment will be done by one of the reviewers and checked by the second reviewer. Any discrepancies in judgement will be discussed until consensus is reached.” 

9. Please consider (not mandatory) stakeholder engagement in the review

Thank you for this suggestion, certainly a worthwhile exercise. While it will not formally part of this scoping review, we will be gathering stakeholder input through two round table events on this topic. Both the scoping review and data collected from the round table events will feed into a practical guidance document for researchers in environment and health research. We are considering the scoping at this point as a stand-alone academic piece to base further discussions on. 

General

1. Please define ethical decisionmaking and research areas in S1 file

We can perhaps contextualise the results of the review within a more specific theoretical framework for decision-making. However, the authors wanted to avoid making too many assumptions before analysing the data. Therefore, within the context of the protocol we are keeping a broad definition. 

The following definitions have been added in the S1 file: 

Definition “ethical decision-making”: No specific theoretical model for ethical decision-making is assumed within the context of this scoping review protocol. It is loosely defined as the process of translating a set of heuristics rooted in the subject´s (within this protocol usually referring to a researcher) ethical values, moral principles, and experience, into actions when faced with an ethical dilemma, taking into consideration external situational factors.

Definition “research areas”: geographical extent of the location in which a research project is being conducted. Specifically, when pertaining to drone research; the geographical extent over which drone flights are being conducted.

2. Please include a list of abbreviations

A list of abbreviations is added at the end of the protocol, after the section "Changes to the protocol" and before the "Acknowledgements" section. 

---

## [Decision Letter · Decision Letter 1]

18 Dec 2023

Ethical considerations related to drone use for environment and health research: a scoping review protocol

PONE-D-23-15407R1

Dear Dr. Hoek Spaans,

We’re pleased to inform you that your manuscript has been judged scientifically suitable for publication and will be formally accepted for publication once it meets all outstanding technical requirements.

Kind regards,

Sathishkumar Veerappampalayam Easwaramoorthy

Academic Editor

PLOS ONE

Additional Editor Comments (optional):

Reviewers' comments:

Reviewer's Responses to Questions

**Comments to the Author**

1. Does the manuscript provide a valid rationale for the proposed study, with clearly identified and justified research questions?

Reviewer #1: Yes

Reviewer #2: Yes

2. Is the protocol technically sound and planned in a manner that will lead to a meaningful outcome and allow testing the stated hypotheses?

Reviewer #1: Yes

Reviewer #2: Yes

3. Is the methodology feasible and described in sufficient detail to allow the work to be replicable?

Reviewer #1: Yes

Reviewer #2: Yes

4. Have the authors described where all data underlying the findings will be made available when the study is complete?

Reviewer #1: Yes

Reviewer #2: Yes

5. Is the manuscript presented in an intelligible fashion and written in standard English?

Reviewer #1: Yes

Reviewer #2: Yes

6. Review Comments to the Author

You may also provide optional suggestions and comments to authors that they might find helpful in planning their study.

Reviewer #1: After the review of the text I consider the work is more educational. However I suggest the authors have to avoid the verbalism when they write.

Reviewer #2: The authors have comprehensively answered all my comments and edited the manuscript accordingly. A suggestion is to transfer the contents of the 'acknowledgement' section to 'author contribution'.

7. PLOS authors have the option to publish the peer review history of their article (what does this mean?). If published, this will include your full peer review and any attached files.

Reviewer #1: **Yes: **Dr Ioannis Vogiatzis

Reviewer #2: **Yes: **Prachi Pundir

---

## [Editor Report · Acceptance letter]

22 Jan 2024

PONE-D-23-15407R1 

PLOS ONE

Dear Dr. Hoek Spaans, 

I'm pleased to inform you that your manuscript has been deemed suitable for publication in PLOS ONE. Congratulations! Your manuscript is now being handed over to our production team.

Kind regards, 

on behalf of

Dr. Sathishkumar Veerappampalayam Easwaramoorthy 

Academic Editor

PLOS ONE